# Identifying Prenatal and Postnatal Determinants of Infant Growth: A Structural Equation Modelling Based Cohort Analysis

**DOI:** 10.3390/ijerph181910265

**Published:** 2021-09-29

**Authors:** Kelly Morgan, Shang-Ming Zhou, Rebecca Hill, Ronan A. Lyons, Shantini Paranjothy, Sinead T. Brophy

**Affiliations:** 1School of Social Sciences, Cardiff University, Cardiff CF14 4YS, UK; morgank22@cardiff.ac.uk; 2Centre for Health Technology, Faculty of Health, University of Plymouth, Plymouth PL4 8AA, UK; 3WHO Collaborating Centre on Investment for Health and Well-Being, Public Health Wales, Cardiff CF10 4BZ, UK; rebecca.hill@wales.nhs.uk; 4Health Data Research UK, Institute of Life Science, Swansea University, Swansea SA2 8PP, UK; r.a.lyons@swansea.ac.uk (R.A.L.); s.brophy@swansea.ac.uk (S.T.B.); 5School of Medicine and Dentistry, University of Aberdeen, Aberdeen AB25 2ZD, UK; shantini.paranjothy@abdn.ac.uk

**Keywords:** infant growth, structural equation modelling, pregnancy, public health, physical activity, paediatrics, obesity, postnatal development

## Abstract

Background: The growth and maturation of infants reflect their overall health and nutritional status. The purpose of this study is to examine the associations of prenatal and early postnatal factors with infant growth (IG). Methods: A data-driven model was constructed by structural equation modelling to examine the relationships between pre- and early postnatal environmental factors and IG at age 12 months. The IG was a latent variable created from infant weight and waist circumference. Data were obtained on 274 mother–child pairs during pregnancy and the postnatal periods. Results: Maternal pre-pregnancy BMI emerged as an important predictor of IG with both direct and indirect (mediated through infant birth weight) effects. Infants who gained more weight from birth to 6 months and consumed starchy foods daily at age 12 months, were more likely to be larger by age 12 months. Infant physical activity (PA) levels also emerged as a determinant. The constructed model provided a reasonable fit (*χ*^2^ (11) = 21.5, *p* < 0.05; RMSEA = 0.07; CFI = 0.94; SRMR = 0.05) to the data with significant pathways for all examined variables. Conclusion: Promoting healthy weight amongst women of child bearing age is important in preventing childhood obesity, and increasing daily infant PA is as important as a healthy infant diet.

## 1. Introduction

The growth and maturation of infants reflect their overall health and nutritional status. The first 1000 days of life, including gestation and the first two years of age, is a critical period of infant growth that may shape the adulthood health and well-being of individuals [1,2]. Understanding the factors that contribute to infant growth in terms of weights and sizes allows the early detection of childhood health problems, such as childhood obesity, pathological deviations (e.g., short stature due to malnutrition and infection [3,4], poor weight gain due to genetic, endocrine and metabolic disorders [5,6,7,8]) and informs successful prevention strategies [1,5]. Emerging evidence shows that obesity is developed through complex interactions between genetic, physiological, behavioural and environmental factors rather than a single factor [9,10,11].

However, despite the identification of several determinants in the development of childhood obesity including biological, social, and behavioral factors [6], the majority of existing analyses focused on studying the influences of either prenatal exposures or postnatal exposures on infant growth and weight. Few studies have investigated the factors of joint prenatal and postnatal exposures and their interactions which may make an impact on infant growth [7]. Both preventative and treatment interventions are yet to tackle this problem. One difficulty is the collection of comprehensive data encompassing a wide range of prenatal and postnatal exposures with birth outcomes and infant growth measurements [8]. Another difficulty lies in the identification of true causal pathways due to the complexity of the interactions between numerous influential variables [9]. Because a large number of variables/factors are usually involved, it would be infeasible to manually identify such complex casual relations by domain experts. The aim of the present study is to provide a more comprehensive analysis of both prenatal and postnatal influences on infant growth via a data-driven approach This will be achieved by utilizing an exploratory Structural equation modelling (SEM) approach to identify and evaluate predictors of infant weight and waist circumference at age 12 months, and characterise the pathway of the impact of these predictors on infant growth.

As an emerging approach to multivariate analysis of causal relations amongst observed and unobserved (latent) variables, SEM [10] offers a good analytics tool to explore the impact of the interactions between observed and unobserved risk factors within the family environment on infant growth (IG) [11]. Indeed, developing such an SEM model with the use of extensive measures from both prenatal and postnatal exposures will enable the synergistic or additive effects of factors to be established, and help with identifying the contribution of each factor to the IG and development. Particularly, it will help identify potentially modifiable factors amenable to intervention and the main targeted components in order to inform the development of effective intervention strategies. Our approach is underpinned by the Barker hypothesis [12] stating that intrauterine growth retardation, low birth weight, and premature birth have a causal relationship to the origins of hypertension, coronary heart disease, and non-insulin-dependent diabetes, in middle age.

## 2. Materials and Methods

### 2.1. Study Population

Mother–child pairs who had participated in a prospective birth cohort study—Growing Up in Wales: Environments for Healthy Living (EHL) [8,13], were included for the present study. Pregnant women were recruited from South Wales, United Kingdom, between November 2009 and March 2013. Briefly, after providing informed written consents, pregnant women over age 16 years old, receiving prenatal care within the Swansea Bay University Health Board (former Abertawe Bro Morgannwg University Health Board) area south of Wales, UK, were recruited to have face-to-face interviews at antenatal appointments. A one-off data collection visit was conducted during pregnancy and subsequent 12 month postnatal follow-up visit. The collected anthropometric, demographic, and questionnaire data included family characteristics, dietary intake, physical activity, and family-related variables. Postnatal data were also collected from both participant maternity notes and the parent-held child health records at the 12 month follow-up (including infant weight, length, and infant feeding method). With the consent, prenatal records, postnatal notes and routine electronic medical records were accessed via the Secured Anonymised Information Linkage Databank [14]. In total, 274 mother–child pairs were recruited for this study. A full description of participant involvement has been published elsewhere [8]. Inclusion criteria for the present study were: (1) availability of maternal demographics and anthropometric measures; (2) a singleton pregnancy; (3) no record of gestational diabetes mellitus (GDM); GDM increases the risk of adverse pregnancy and neonatal outcomes, and the risk of obesity in later life of both the mother and the child [15]. Treatment for GDM and potential exposure to diabetes in the uterus may introduce bias and unmeasured confounding variables to the whole sample in the present study. For this reason, women with GDM were excluded; and 4) no chronic health condition in the infant. Figure 1 shows how the inclusion criteria were applied to derive the sample for this analysis.

### 2.2. Studied Variables

Appendix A (see online Appendix A) provides an in-depth account of studied variables outlining details of assessment and time points of collection. Questionnaire data provided information on maternal age at delivery, annual household income, monthly allowance payments, marital status, ethnicity, highest level of education and current employment status. Data were also collected on frequency of maternal smoke exposure, smoking status and alcohol consumption during pregnancy and maternal mental health (based on a self-reported Kessler 6 score). Maternal pre-pregnancy BMI was recorded by a midwife at 12 weeks gestation, and obtained from antenatal records. A seven-day, self-reported, diet diary provided average daily consumption of carbohydrates, protein, fat, saturated fat and fibre during pregnancy. Objective seven-day physical activity readings (during pregnancy (predominantly during the second or third trimester) and at 12-month follow-up) were recorded using a validated accelerometer—triaxial GENEActiv accelerometer [16]. The GENEActiv was mounted underneath clothes with good skin contacts on the wrist (mothers) or ankle (infants [17,18]) to collect the movements for each second at 100-Hz frequency. A more detailed explanation of GENEActiv accelerometer type, use and captured data have been described elsewhere [16]. A physical activity event detection algorithm [17] was used to identify the wear and non-wear events.

Maternal height, weight, skinfold thickness (biceps, triceps, subscapular and suprailiac) and mid-arm circumference were measured during pregnancy. The mean gestation of women was 24.4 weeks (SD 1.6, range 6.3–35.1) when data collection occurred.

Birth weight, gestational age and Apgar scores (at 1 and 5 min) were obtained from medical notes recorded by a health visitor. These pieces of information were also available from clinical records. Average weight gain per week (from birth to 6 months) was calculated by taking a subsequent measure of weight from clinical records. At 12 months postpartum, follow-up questionnaires provided; details on any changes in maternal living circumstances, infant feeding practices (breastfeeding initiation and duration, weaning age and a 7-day food frequency questionnaire), sleep patterns, developmental stages (age of crawling and walking), physical activity patterns and TV viewing time. An experienced and trained researcher obtained infant anthropometric measures which also included length, and mid-arm circumference (as measured in the mother). Using such a data-driven approach, our analyses can produce a relatively simple model displaying causal pathways. Some studies have indicated the associations of infant weight and waist circumference with adverse health outcomes, such as future obesity and cardiovascular risk [16,17,18]. A long-term study conducted by Schmidt et al. [19] suggested that waist circumference, rather than the commonly used BMI measure, is the best clinical measure to predict a child’s risk for cardiovascular disease and diabetes later in life.

Infants vary in size and shape as much as adults do. Different anthropometric parameters were developed [20], weight and waist circumference are commonly used. In this paper, the study outcome, infant size, was a latent variable composed of a weight and waist circumference measure, obtained at the 12-month follow-up. Our model aimed to identify factors which were associated with a heavier infant with a larger waist circumference.

### 2.3. Statistical Analyses

A two-step statistical analyses approach was adopted for the present study: first, a series of regression analyses (enabling model specification, see Appendix A), and second, a SEM approach. The SEM was carried out using STATA software (version 13.1) in order to examine the pathways between pre- and early post-natal influences and infant size at age 12 months. After testing the exploratory SEM model (Appendix A), the approach was strengthened to include only statistically significant pathways (*p* ≤ 0.05) (Figure 2). To examine how well the sample’s variance-covariance data fitted the SEM model, we used four fitting indexes: the Chi-square test (*X*^2^), the Root Mean Square Error of Approximation (RMSEA), the Standardized Root Mean Square Residual (SRMR) and the Comparative Fit Index (CFI), and followed the cut-offs for good fitting models suggested by Hu et al. [21]. Maximum Likelihood of Missing Values (MLMV) was performed to enable subsequent analyses on a full data set, assuming missing at random (MAR). Subsequent models also incorporated gender.

## 3. Results

### 3.1. Description of the Data

Table 1 displays explanatory variables (prenatal and postnatal) for mothers and infants.

Weight, waist circumference and skinfold measurements at age 12 months were available for 270 (141 boys and 129 girls), 225 (122 boys and 103 girls) and 202 (111 boys and 91 girls) infants respectively. Eighteen per cent (n = 49) of the sample did not have an outcome variable available and were excluded from the main SEM analyses.

Comparing maternal characteristics between those mothers who did (n = 274) and did not (n = 269) participate in the 12 month visit, mothers who did participate were; more likely to be older (*p* < 0.0001) and to have received higher education (0.015, 95%CI: 0.1 to 0.2), less likely to smoke during pregnancy (Odds: 0.11, 95% CI: 0.03 to 0.2) and be primiparous (Odds: 0.13, 95% CI: 0.06 to 0.2). No significant differences were shown between ethnicity groups, pre-pregnancy BMI values, the proportion of mothers in full-time employment or those reporting an annual household income of less than £15,000.

Objective physical activity (measured by accelerometer) were available for 63 infants (25.5%) of which we examined the relationships between hours spent sedentary and intensity of time spent active with infant size. Although advised to wear the device for 7 days, only 6 days with 24 h data were used (as the home visit took place during the day, the first day did not provide a full 24 h). The average wear time was 4.5 days with 55% of infants wearing the device for the full 6 days. On average infants were shown to be sedentary for 10.3 h per day (range 1.5–18.2 h). Of the infants with physical activity data, 49% had taken their first steps. No significant differences in the number of hours spent sedentary or the intensity of active periods were shown between the two groups. Examining the data for each gender, females spent less time being sedentary (mean 9.13 h (SD 5.2) vs. Mean 10.05 (SD 5.5)) and males showed a higher intensity during active periods (mean SVM 266,172 (SD 148,015) vs. 255,825 (SD 133,016)) however, neither finding reached statistical significance.

### 3.2. Direct, Indirect and Total Effects from the SEM

For the ease of interpretation, Figure 2 displays the final model with accompanying standardised pathways affecting infant size (the standardised estimates are provided in Table 2). Descriptive accounts for each variable are located at online Appendix A. All path coefficients in the model were statistically significant at the level *p* < 0.05, signifying that each variable made a meaningful contribution to the model. The relationship noted between maternal weight (pre-pregnancy BMI) and infant size was of modest size (β = 0.12). An indirect effect of pre-pregnancy BMI on infant size was evident with infant birth weight acting as a mediator (β = 0.08). Self-reported infant consumption of carbonated drinks (β = −0.18) and lower levels of infant play (β = −0.14) were related to smaller infant size. Weight gain was not related to birth weight or any infant feeding measure. Notably, no pathway involving maternal physical activity or maternal diet was apparent within our final model.

The final model displays an improved goodness of fit (Table 3); however, the significant chi-square value does not meet goodness of fit criteria. Comparing the other fit indices to generally accepted cut-offs, the RMSEA and SRMR appear to indicate reasonable model fit. The CFI value at 0.94 falls slightly below the newly devised threshold (acceptable threshold values greater than 0.95 [22]) but exceeds the older threshold of 0.90 [21].

### 3.3. Modelling by Gender and Imputing Data

Modelling by gender revealed the modest path coefficients, as the significance levels of most pathways dropped (*p* ≥ 0.1). Only carbohydrate consumption, for both genders and birth weight for boys (*p* < 0.05) remained significant predictors within the model. The goodness of fit indices of SD and SRMR (girls model only) reveals a poor fit for both genders whilst further analyses showed that infant carbohydrate consumption was significantly different between the two groups (X^2^ = 4.02, *p* < 0.05). Using MLMV with all 225 mother–child pairs produced a similar model as that of the true data, demonstrating similar path coefficients and negative effects of infant play and carbonated drink consumption. Effect sizes for each variable within the latent construct remain unchanged. A noticeable difference is that infant-play no longer demonstrates a significant association with infant size. Table 2 provides standardised estimates for the modified, full and gender-specific models.

## 4. Discussion

### 4.1. Prenatal Exposures

Our model revealed that some maternal factors, such as age, ethnicity, education and income did not work significantly to predict early infant size, while the maternal pre-pregnancy BMI was an important factor affecting the infant size in both direct and indirect causal effects; the latter was mediated through infant birth weight. This finding is consistent with an existing study which evidenced that maternal BMI has a direct influence on infant and early childhood BMI growth [23]. In line with the existing evidence [24], our finding suggests that infant size is not entirely attributed to exposures after birth, raising the concern about the current interventions and policies focusing solely on postnatal factors [25]. The association between maternal pre-pregnancy BMI and offspring birth weight has previously been attributed to processes underlying foetal programming as described by the Barker hypothesis [12]. Therefore, infants born larger (e.g., heavier rather than longer) are more likely to remain large (heavy with a larger waist size) throughout infancy and later years.

Demonstrating both a direct and indirect causal impact of maternal pre-pregnancy BMI on infant size, the observations from the present study are also indicative of a synergistic effect. It is possible that an underlying mechanism between genetic factors and the family environment ensues with intergenerational similarities manifesting through foetal development and similar maternal–offspring lifestyles. Previous studies have shown strong associations between maternal [26,27] food consumption, parental food consumption [27], and offspring diet in the early years. Despite finding no influence of maternal dietary intake (during pregnancy or at 12 months) on infant dietary patterns within our study, our model provides the justification for further exploration. This finding is encouraging because it shows that measuring a single aspect of maternal health can enhance our understanding of future offspring size. In light of the high number of observations whereby mothers and infants reported undesirable dietary intakes at 12-months (e.g., a high number of crisps, biscuits and chocolate/sweets), future research is warranted to examine the infant diet. Such research is required to assess whether the same food products are shared between mothers and infants or whether infants are given specific infant brands. This would help shed greater light on the observed findings and the role of the shared familial environment.

Furthermore, this finding suggests that future interventions and policies should focus on reducing pre-pregnancy BMI in order to have an impact on the size of future offspring.

### 4.2. Postnatal Growth

Infant weight gain emerged as a significant predictor of infant size, independent of all other variables within our model. Due to the availability of weight measures and the level of missing data, we were unable to comment on specific phases or patterns of weight gain throughout infancy. Subsequently, our weight gain measure is a crude evaluation and does not reflect the process by which infants gained weight [28]. Nonetheless, our finding is supported by an abundance of literature detailing the significant association between increasing weight gain in early life and subsequent obesity risk [5,29]. Given that our measure of infant size is a combination of infant weight and infant waist circumference, existing research which describes lower levels of central subcutaneous fat amongst exclusively breastfed children at ages 1–2 years further supports our findings [30]. Previous research detailing the effects of increasing infant length on weight gain [31] could not explain our findings, as our model was inclusive of infant length at age 12 months.

### 4.3. Postnatal Exposures

Two infant dietary measures turned out to be important predictors of infant size. Firstly, our observed association between daily consumption of starchy foods and larger infant size could portray an effect of simple carbohydrates on increasing weight gain. It is speculated that infants who were introduced to a high quantity of carbohydrates too early in life are more likely to boost insulin secretion with recursive metabolic adaptations, and an increasing susceptibility to elevated weight gain and obesity later on in life [32].

Our model examined smaller sizes amongst those infants consuming carbonated drinks (one or more days a week) when compared to those reporting no consumption. With approximately 43% of an infant’s diet attributed to beverage intake at age 1 year [33], it is possible that the consumption of carbonated drinks creates a ‘fullness effect’ in infants and in turn reduces the intake of milk and other nutritious food items. Recent findings from analyses of NHANES (National Health and Nutrition Examination Survey) data, have reported a displacement of milk with carbonated drinks in children aged 1 to 5 years [34]. Other studies have examined sweet drink consumption amongst a sample of preschool children aged 2–5 [35] and 2–3 years [36]. Whereas Newby and colleagues [35] found no significant associations between beverage consumption (soda, fruit juices and milk) and weight gain amongst low-income children, a retrospective study reported increasing weight gain amongst infants consuming 1 or more sweet drinks (soda, fruit drinks, juices and other sweetened beverages) per day [36]. However, associations between artificially sweetened beverage/sugar-sweetened beverage consumptions and risk of obesity were found for all children, not just low-income children [37]. With earlier findings showing that food preferences in children as young as age 2 are highly predictive of food choices in later childhood [33,38], it is important to educate mothers on healthy diet options for the infant, addressing both food and beverage choices.

No statistically significant differences in the number of hours spent sedentary or the intensity of active periods were found between females and males. Lower levels of infant physical activity were also associated with smaller infant size, that is, playing physically active games less than once a day was associated with lower weight and a smaller waist circumference. The observed effect size was comparative to the effect shown of starchy foods but in a counteractive fashion, revealing an inverse effect. A potential explanation for this association lends to previous research on bone properties. Research involving both children and adolescents has shown the benefits of physical activity on bone density [39], which could explain the observed smaller sizes amongst less active infants. Total time spent partaking in weight-bearing activities, self-reported sports and play activities have been significantly related to bone mineral density amongst children as young as 5-years [39]. It is also possible that this association could portray differences in infant body composition [40], with lower levels of muscle mass in those infants playing less often resulting in lower weights. Nonetheless, even using a weak proxy measure, our finding demonstrates that physical activity in children as young as 1 year of age, contributes to size in early life and should not be overlooked rather than solely focusing on dietary factors. Increasing infant activity could provide an avenue for combating adverse effects triggered by dietary factors.

### 4.4. Strengths and Limitations

The present study innovatively adopts an SEM approach to examine longitudinal data with prenatal and postnatal variables measured from early pregnancy to 1 year postpartum, providing a platform for analysis based on a life course approach. Using a data-driven approach, our analyses produced a relatively simple model displaying causal pathways of infant size. Modelling infant size as a latent variable, we were able to account for both increased weight and larger waist circumference, evading problems associated with collinearity. The majority of infant size did, however, consist of infant weight.

There are however some limitations with this analysis. First, due to the incompleteness of data, our model was based on a significantly reduced sample of the total study population. This could be a contributing factor to the model’s goodness of fit estimations and also limited our ability to produce gender specific models. Nonetheless, the observed pathways were all significant and our model does provide a greater understanding of pre- and postnatal interactions. The high variation observed amongst infant sizes and measured factors also warrant the need for larger datasets to strengthen our observations and identify further pathways. Second, our studied population was predominantly white and well educated, with two thirds of women presenting as first time mothers at study enrolment. These characteristics limit the generalization of our findings to the wider population with diverse ethnic groups and lower education levels. Third, as with all cohort studies, we cannot rule out any potential effects of unmeasured variables within our model. Throughout our study design, we were unable to account for factors relating to the non-shared family environment, such as interactions with siblings and the influence of childcare arrangements. Additionally, no consideration was given to paternal factors. It is therefore possible that infant size is mediated by other factors which are not included within our study design.

Moreover, the reliance on mothers to provide all dietary data relating to child-feeding practices, infant dietary content, and their own diet during pregnancy and at 12 months is subject to limitations. Even though every effort was made to encourage mothers to record dietary data on a day-to-day basis, mothers often reported recall problems and completion of their diet diary over several days as opposed to daily. Within the infant dietary data, it was difficult to distinguish between infant specific items; therefore, it is possible that infants were consuming specific tailored brands and not adult products as assumed in the present study. As a result, our finding of smaller sizes with higher consumptions of carbonated drinks could be indicative of infants consuming carbonated drinks suitably lowered in sugar; however, this is speculative. Moreover, our ability to comment on the causal pathways involving infant dietary predictors is somewhat limited as infant dietary measures were obtained at the same time point as our study outcome. Finally, our measures of size did not look at body composition. High carbohydrates are associated with larger (overweight infants) but higher physical activity is associated with larger (more muscle) infants. This is speculative and is not based on body composition measurements. In addition, our measurements of infant physical activities did not consider some situations, such as the infants being picked up or carried.

### 4.5. Implications

The results of the present study suggest that future efforts aiming to prevent childhood obesity should focus on promoting healthy weights amongst women of childbearing age, by encouraging healthy diets and active lifestyles. In addition, public health programmes, such as the Healthy Child Programme, should include interventions that facilitate the adoption of healthy family lifestyle approaches. Further longitudinal, large-scale studies, involving in-depth dietary and physical activity assessments are needed to confirm our findings.

## 5. Conclusions

This study presented an SEM approach to evaluate the impact of prenatal and early postnatal factors as predictors of growth of infants at age 12 months. The research findings supported that promoting healthy weights amongst women of childbearing age is important in preventing childhood obesity, and increasing daily infant physical activity is just as important as the emphasis placed on the need for a healthy infant diet.

Some machine learning methods have emerged to identify the causal features from the large dataset [41,42]. These could be a new way forward to analyse the current dataset for future work.

## Figures and Tables

**Figure 1 ijerph-18-10265-f001:**
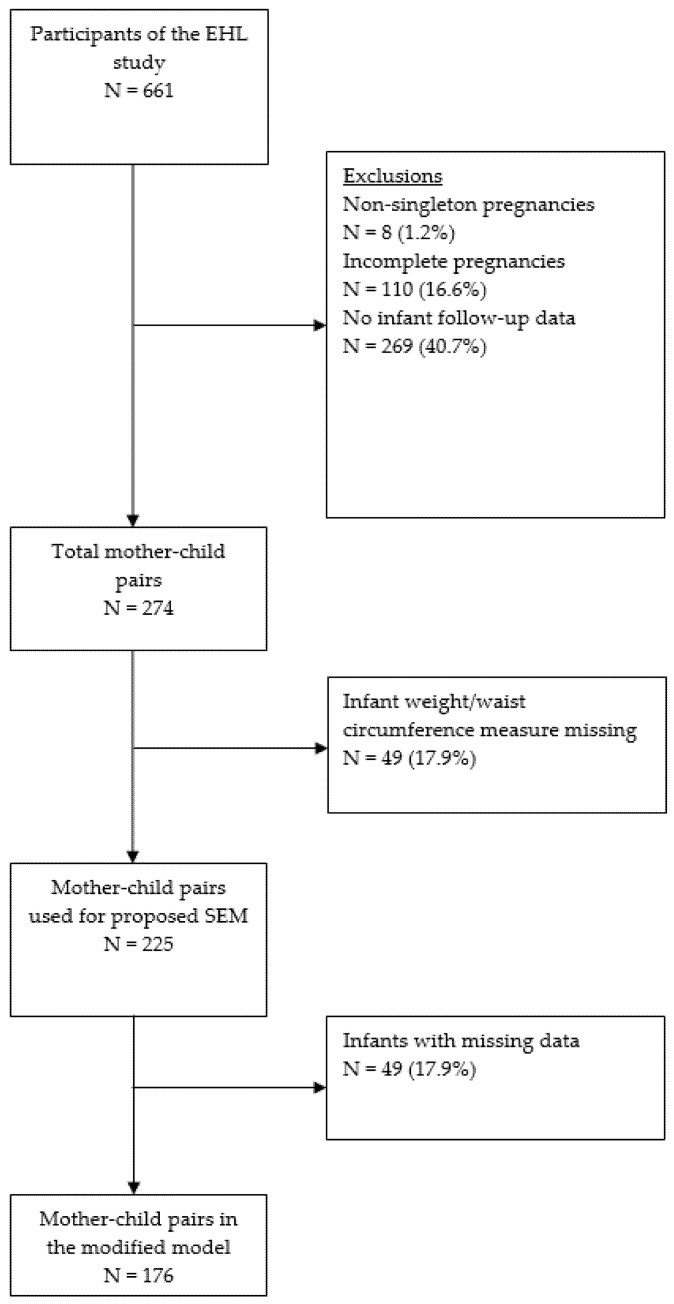
Flow diagram depicting the exclusion of mother–child pairs in the present study.

**Figure 2 ijerph-18-10265-f002:**
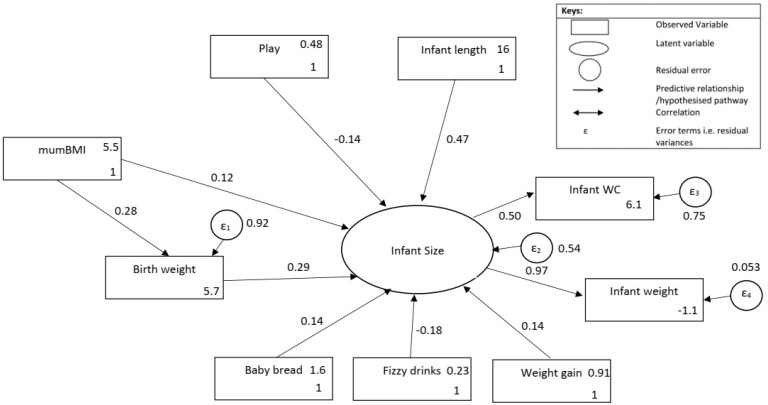
Modified structural equation model displaying standardised coefficients (*χ*^2^ (11) = 21.5, *p* < 0.05; RMSEA = 0.07; CFI = 0.94; SRMR = 0.05). *mumBMI* (mother’s BMI at 12 weeks gestation), *play* (infant plays with parent or physically active games daily or less often), *infantlength* (length at age 12 months), *babybread* (infant consumes carbohydrates daily or less often), *fizzydrinks* (infant consumes carbonated drinks daily or less often), *weightgain* (infant average weekly weight gain (g) from birth to 6 months), *infantsize* (factor of infant waist circumference (infant wc) and infant weight, both measured at age 12 months).

**Table 1 ijerph-18-10265-t001:** Characteristics of the total study population of mother–child pairs.

	Samples Available	Mean (SD) or %
	N	
**Maternal characteristics**		
**Socio-demographics during pregnancy**		
ge at delivery (years)	264	30.9 (5.6)
White ethnicity	270	89.5%
First or higher degree attained	201	50.2%
Household income <£15,000	238	21.0%
Full time employment	267	43.1%
Smoked during pregnancy	259	15.8%
**Reproductive factors**		
Pre-pregnancy BMI (kg/m^2^)	274	24.4 (4.7)
Healthy weight	149	54.4%
Overweight	89	32.5%
Obese	36	13.1%
Number of children <1	274	83.10%
**Socio-demographics at 12 months**		
Household income <£15,000	226	20.8%
Full time employment	223	25.6%
**Infant characteristics**		
Male	122	52.6%
Mean birth weight (kg)	274	3.4 (0.5)
% <2.5 kg	5	1.5%
% >4.0 kg	37	13.5%
Mean gestation (weeks)	225	39.6 (1.5)
% born <37 weeks	8	3.6
% born ≥42 weeks	12	5.3
Mean weight gain 0–6 months (per week)	187	0.14 (0.04)
% Breastfed at birth	228	84.2%
% Breastfed at age 12 months	227	20.4%
Mean Weight at 12 months (kg)	270	9.8 (1.2)

**Table 2 ijerph-18-10265-t002:** Standardised estimates for the modified, full and gender-specific models.

Parameter Estimate	Modified Model		Full Model	Gender Specific	
	Unstandardised	Standardised		Males	Females
N	176	176	225	97	79
**Structural Model**					
Pre-pregnancy BMI ≥ birth weight	0.03 (0.01)	0.28 (0.07) ***	0.26 (0.06) ***	0.26 (0.1) *	0.31 (0.1) *
Birth weight ≥ Size	1.3 (0.34)	0.29 (0.07) ***	0.32 (0.05) ***	0.36 (0.09) *	0.22 (0.1) *
Infant length ≥ Size	0.19 (0.05)	0.47 (0.06) ***	0.38 (0.06) ***	0.52 (0.1) *	0.4 (0.1) *
Weight gain ≥ Size	0.73 (0.27)	0.14 (0.07) **	0.22 (0.06) ***	0.12 (0.1) *	0.12 (0.1)
Play ≥ Size	−0.72 (0.36)	−0.14 (0.08) **	-0.09 (0.06)	−0.14 (0.15) *	−0.15 (0.1) *
Infant carbohydrate consumption ≥ Size	0.65 (0.31)	0.14 (0.08) **	0.15 (0.06) *	0.28 (0.01) **	0.01 (0.09) *
Infant fizzy drink consumption ≥ Size	−1.66(0.64)	−0.18 (0.08) ***	−0.16 (0.06) *	−0.17 (0.1) *	−0.25 (0.1) *
Pre-pregnancy BMI ≥ Size	0.06 (0.03)	0.12 (0.08) *	0.18 (0.07) ***	0.1 (0.1)	0.15 (0.1) *
**Measurement model**					
Size ≥ Infant waist circumference	1	0.5 (0.09) ***	0.54 (1.10) ***	0.49 (0.08) ***	0.61 (0.09) ***
Size ≥ Infant weight	0.52 (0.10)	0.97 (0.12) ***	0.97 (0.83) ***	0.89 (0.11) ***	0.90 (0.12) ***

Unless stated all values are standardised with standard error in Parentheses, with *p* < 0.1 (*), *p* < 0.05 (**), and *p* < 0.001 (***); Fit indices- Full model: *X*^2^ (11) = 22.5, *p* < 0.05; RMSEA = 0.07; CFI = 0.95; Gender-specific model: Males SRMR = 0.08, Coefficient Determinant (CD) = 0.48; Females SRMR = 0.14, CD = 0.46.

**Table 3 ijerph-18-10265-t003:** Goodness of fit statistics for the proposed and modified model with acceptable thresholds.

Fit Indexes	Proposed Model	Modified Model	Good Fit Level
*X*^2^ (df)	44.49 (15) **	21.5(11) *	Close to 0 (*p* < 0.05)
RMSEA	0.11	0.07	<0.07
CFI	0.84	0.94	>0.90
SRMR	0.06	0.05	<0.08

* *p* < 0.05; ** *p* < 0.001.

## Data Availability

The use of raw data needs approval in advance. Appendix A were provided.

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
