# Peer review of "Identifying Prenatal and Postnatal Determinants of Infant Growth: A Structural Equation Modelling Based Cohort Analysis"

_ijerph, 2021, doi:10.3390/ijerph181910265_

Round 1

Reviewer 1 Report

The authors have successfully addressed all of my queries up satisfaction. I recommend that the paper be accepted as it is for publication.

Reviewer 2 Report

Thank you for your revisions. I was unable to see your comments but have re-read the paper and have a few minor suggestions. Additionally, please review the paper for additional areas that need English editing.

  • Line 16 – change to “The purpose of this study…”
  • Lines 40-42 – please reword, these sentences are not clear and do not make sense.
  • Line 49 – please add “Which may make an impact…”
  • Line 50 – change to “this problem.”
  • Line 41 – remove “a” from …is the collection of comprehensive data encompassing…
  • Line 58 – add “a” to ..via a data-driven approach..
  • Line 64 = change to “A long-term study…”
  • Line 71 – You need to define this acronym at first use in the first paragraph and revise throughout the intro as you have spelled out infant growth several times.
  • Line 62-67 – While this information is good, it does not flow well where it is currently written especially since the next paragraph goes back to discuss SEM. I would suggest moving it to the methods where you begin to speak about this in line 140.
  • Line 121-122 – please remove the word accelerometer once in this sentence.
  • Line 123 – were they instructed to wear it on a certain wrist or ankle?
  • Line 137 – you have this above, remove in on spot.
  • Line 186-187 – please use the actual name instead of inserting the # of the reference. Also, this should be in the methods instead of the results
  • Line 205 – please clarify that this is the self-reported variable or the accelerometer data, additionally this variable should be described in the methods section.
  • Please add to your discussion of the impact of physical activity about your results that there were no differences in objective physical activity between the two groups

Author Response

This manuscript is a resubmission of an earlier submission. The following is a list of the peer review reports and author responses from that submission.

Round 1

Reviewer 1 Report

This research study made an effort to investigate the associations between prenatal and postnatal factors and infant growth at 12 months. There have been few cohort studies aiming at these associations. This paper is well written to present a two-step statistical modeling analysis approach. Some notable strengths to this approach include using a large number of variables, comprehensive real world survey data collected to represent mothers and infants, research results are meaningful and can bring good implications in public health. However, the paper needs revisions.

First this paper claimed the contribution of this study by using a "data-driven" approach. The authors need to highlight why the called "data-driven" approach is necessary and important to this study.

Table S5 is very helpful to understanding the contribution of the data analysis approach, it should be put into the Discussion section about strengths and limitations of the paper.

Table 1 and Table S3 seem overlapping. I suggest the authors to only use one table of them.

The statistical inference adopted in this research is based on series of regression analyses and structural equation modelling. There are some machine learning nonlinear models that attempt to perform nonlinear inference or identify the causal features from the dataset. These could be a new way forward to analyse the current dataset for future work. For example, see the work of Rodríguez-Rodríguez et al, A Comparison of Feature Selection and Forecasting Machine Learning Algorithms for Predicting Glycaemia in Type 1 Diabetes Mellitus, Appl. Sci. 2021, and Koh et al, Deep Temporal Convolution Network for Time Series Classification, Sensors 2021. The recent techniques proposed in these work can serve as a platform for (causal) data analysis.

Reviewer 2 Report

This manuscript describes a method for applying structural equation modeling to pregnancy and postnatal data to predict infant size at 12 months, an approach that is posited as an improvement upon multivariable regression in terms of dealing with interactions between risk factors and identifying the contribution of each factor to the outcome. They build their model using data from the Growing Up in Wales EHL study, which enrolled more than 600 women; however, they only ended up with 176 mother-child pairs in their analytic sample, which limited their ability to detect statistically significant findings in sex-stratified analyses and the generalizability of their results.

The manuscript is well written overall, but certain important points are not as clearly described as they need to be, and there appear to be some inconsistencies.  Their conclusions are also somewhat overstated, especially the section on prenatal influences.

Introduction: It is unclear whether the focus of the paper is on growth or obesity.  The former is generally considered to be desired (e.g., vs. stunting) while the latter is to be avoided.  

What is the evidence that the latent variable that includes infant weight and waist circumference is associated with adverse health outcomes (the purported public health motivation of the study)?  Is it a measure of growth (positive) or obesity (negative)? 

Flow chart and Table 2: The flow chart indicates that 225 mother-child pairs were used for the creation of the full SEM and 176 were used for the modified SEM.  In Table 2, the n is 225 for the full model and 178 for the modified model--not the same.  The number of male + female participants in the stratified models adds up to 174--different yet again!  Also, how could the full model (full = more variables) have a higher n than the modified model (modified = fewer variables) when the lower number reflects missing data?  This is all very confusing.

Table 1: Does this reflect variables that were identified via regression analyses (Table S2--presumably bivariable?  This needs to be specified in the Statistical Analysis section) from the larger list of variables described in Table S1?  This is a complex analysis--please take the time to walk the reader through the methods step by step.  Don't rush!

Some of the variables in Table S1 may not be valid if they were measured only once across such a wide range of gestational ages, e.g., skinfolds, diet, physical activity.  For example, diet measured in the same woman during the first and third trimesters would give wildly different results, especially if she were experiencing morning sickness.  Similarly, physical activity changes markedly across pregnancy.  A single measure at a non-standard time during pregnancy is almost meaningless.  The conclusion that the only prenatal factor to impact infant size is BMI (Discussion) is somewhat disingenuous and needs to be qualified by acknowledging the above-described limitations.  Otherwise, the reader might think that factors such as PA, diet, etc. have been definitively ruled out, which they have not been because of this measurement issue.

The title of Table S3 is confusing.  The final analysis is the modified model, correct?  This only included 176 pairs.  So those 176 should be compared to the 49 who were excluded.  Also, what does the asterisk mean?  If it indicates statistical significance, is this the only variable that was different between the two groups?

Lines 194 and 301ff: The beta coefficient for play is negative.  Doesn't this mean that increased play is associated with decreased size?

Reviewer 3 Report

Thank you for the opportunity to your paper. This paper represents an understudied topic, especially with the inclusion of infant physical activity. There are minor grammatical issues throughout the paper, please review and/or have a colleague review for clarity. I have noted a few below.

Please also consider the following:

  • Line 43 – Remove “that is solely responsible” or add what is responsible for as of now it seems like an incomplete sentence.
  • Line 44-45 – could you add examples in parenthesis of the determinants?
  • Line 48 – would instead say “..and their interactions which impact infant growth.”
  • Line 54-56 – would suggest rewording to “utilizing structural equation modeling to identify…” or something similar, as written it is confusing.
  • Was # of previous pregnancies collected?
  • Line 106 – are these questions validated, especially Kessler?
  • Were accelerometers worn 24 hours per day for mom and baby? Since this special issue is focused on physical activity, please include additional information on where the accelerometers were worn. Has this been used previously? How did you account for infants being picked up/carried, etc.?
  • Line 128 – so was this something your research team developed or has this been used previously? I wasn’t clear from the reference how much it had been used.
  • In the limitations, please address why you think those with higher weights did not have info available and how this might impact results
  • Line 172+ - how did you determine wear time?
  • Line 194 – was the infant play variable a question asked or was this physical activity? I didn’t see the definition of this in the paper or supplementary files
  • Line 208 – I don’t think you have defined CD previously
  • Line 258-259, please expand on how funding is available, I’m not sure this is the case in all countries and/or would have to shift some major political priorities